# The Epigenetic Progenitor Origin of Cancer Reassessed: DNA Methylation Brings Balance to the Stem Force

**Marco Bruschi** [1,2]

1   U981-Molecular Predictors and New Targets in Oncology, INSERM, Gustave Roussy, University Paris-Saclay, 114 rue Édouard Vaillant, 94800 Villejuif, France; marco.bruschi@gustaveroussy.fr; Tel.: +33-(0)1-42-11-64-38
2   Département de Cancérologie de l'Enfant et de l'Adolescent, Gustave Roussy, Université Paris-Saclay, 94800 Villejuif, France

**Abstract:** Cancer initiation and progression toward malignant stages occur as the results of accumulating genetic alterations and epigenetic dysregulation. During the last decade, the development of next generation sequencing (NGS) technologies and the increasing pan-genomic knowledge have revolutionized how we consider the evolving epigenetic landscapes during homeostasis and tumor progression. DNA methylation represents the best studied mark and is considered as a common mechanism of epigenetic regulation in normal homeostasis and cancer. A remarkable amount of work has recently started clarifying the central role played by DNA methylation dynamics on the maintenance of cell identity and on cell fate decisions during the different steps of normal development and tumor evolution. Importantly, a growing number of studies show that DNA methylation is key in the maintenance of adult stemness and in orchestrating commitment in multiple ways. Perturbations of the normal DNA methylation patterns impair the homeostatic balance and can lead to tumor initiation. Therefore, DNA methylation represents an interesting therapeutic target to recover homeostasis in tumor stem cells.

**Keywords:** DNA methylation; adult stem cells; tumorigenesis; tumor stem cells; cancer

## 1. Introduction

According to its most popular definition, epigenetics describes the acquisition of measurable and stably heritable phenotypic traits that do not depend on changes in the DNA sequence itself [1]. Epigenetic control, mediated by the integrative network of histone and DNA covalent modifications, as well as noncoding RNAs, coordinates the cell phenotype and allows genetically identical cells to achieve diverse phenotypic characteristics by modulating the accessibility of different regions of the genome through differential packaging and decoration of the chromatin. However, although the contribution of epigenetic mechanisms is relatively well characterized during embryogenesis [2–6], the precise involvement of those mechanisms in adult stem cell homeostasis has only recently started to be clarified [7]. Of note, much of our current knowledge on the dynamics and function of stem cells actually arises from the study of mechanisms involved in the homeostatic rupture associated with cancer initiation and development. After a long period of mostly being perceived as the simple phenotypic result of sequential mutations in oncogenes and tumor suppressor genes, cancer is nowadays widely recognized to represent the outcome of the combinatorial effect of genetic and epigenetic alterations [8–10]. Alterations in the epigenetic mechanisms are therefore now considered as relevant as genetic mutations in explaining the properties that define cancer cells at the various stages of the disease [10,11]. It comes hereafter as no surprise that the outstanding accomplishments made



during the last few decades in the identification of the genetic changes involved in cancer development have been accompanied by comparable advances in the characterization of the epigenetic control of malignancies [11,12]. This characterization includes (but is not limited to) the role of widespread epigenomic changes, such as the global alteration of DNA methylation profiles, nuclear architecture and chromatin compaction. For multiple reasons, including the relative ease to recover the substrate from different types of biological samples, DNA methylation has so far represented the most well studied epigenetic modification in cancer [13].

DNA methylation plays a plethora of biological roles including X-chromosome inactivation, long-term repression of repetitive and transposable elements, regulation of transcription, and genomic imprinting [14]. Genomic imprinting is a mammalian specific epigenetic mechanism that involves DNA methylation of defined loci resulting in allele-specific methylation and parental-origin-dependent expression of certain genes [15]. DNA methylation is erased during gametogenesis and after fertilization, and re-established *de novo* in the embryo, except at imprinted regions, in which parental patterns are conserved.

In mammals, more than 98% of DNA methylation is found in a CpG context in adult somatic cells, whereas as much as nearly one-quarter of all methylation appears in non-CpG context in embryonic stem cells (ESCs). Detailed information on the genomic distribution and functions of DNA methylation are extensively illustrated elsewhere [16,17]. Briefly, CpG dinucleotides are not randomly distributed across the mammalian genome, but rather they are over-represented within short CpG-rich stretches of DNA known as CpG islands (CGIs). About half of all CGIs coincide with gene promoters, as they contain transcription start sites (TSSs), and the majority of mammalian promoters contain CGIs. The remaining half of CGIs are found within gene bodies or between annotated genes [18]. Importantly, CGIs constitute the most variable regions in terms of DNA methylation extent from one biological context to another, depending on the developmental, physiological or pathological context [19–21]. High extent of DNA methylation at CGIs nearby the TSS or gene enhancers is usually associated with transcriptional repression. Indeed, DNA methylation extent inversely correlates with the abundance of bound transcription factors (TFs). However, the chicken-or-egg debate on whether DNA methylation is prohibited by, or rather impedes the binding of TFs is still not solved in the field [22]. By contrast, DNA methylation is generally depleted at gene bodies of transcriptionally repressed genes in differentiated cells [23]. This correlation, however, does not directly imply that DNA methylation of gene body regions actively promotes the transcription of those genes [16]. Overall, multiple lines of evolutionary evidence suggest that the primary function of DNA methylation in animals consists in the repression of transposable regions. An increasing number of gene promoters might have accumulated CGIs to achieve greater expression diversity of defined genes related to development and cell type-specific functions during the evolution of multicellular organisms [19].

The methylation of DNA is catalyzed by DNA methyltransferases (DNMT) that use *S*-adenosyl-L-methionine (SAM) as a methyl donor [24]. DNMT1 is preferentially recruited to hemimethylated DNA at the replication fork during the S phase of the cell division, and is responsible for replicating methylation patterns to the nascent DNA strand [25]. DNMT1 is therefore commonly considered as the maintenance DNA methyltransferase [26]. DNMT3A, DNMT3B and the catalytically-inactive DNMT3L have increased affinity for unmethylated CpGs and perform *de novo* methylation [27]. Nonetheless, it is well established that DNMT3A and DNMT3B functions can be required for stable epigenetic inheritance in specific contexts [28,29]. It is currently accepted that the erasure of DNA methylation is not merely due to passive replication-dependent dilution of this mark upon successive cell divisions, but is at least in part mediated by the successive enzymatic oxidation of methylcytosine, catalyzed by the ten-eleven translocation proteins TET1, TET2 and TET3 [20,30]. Of note, *TET* genes have proved to act as tumor suppressors in several hematological and solid malignancies [30].

Some alterations associated with DNA methylation are now considered hallmarks of cancer development and currently represent targets for the discovery of biomarkers with diagnostic and

prognostic relevance and for the development of therapeutic strategies. However, the precise dynamics at which these alterations occur and orchestrate tumor initiation, as well as their functional contribution to this process have only recently started to be characterized [31].

Providing selected examples, this review aims at summarizing and discussing the most relevant knowledge on the role of DNA methylation in normal homeostasis and its implications in cancer initiation, by specifically focusing on the control exerted by this epigenetic mark on the dynamics of adult stem/progenitor populations in both these processes. Based on the evidence described, I will then discuss whether some altered DNA methylation patterns in adult stem cells may represent potential targets for recovering the homeostatic balance of the tissue.

## 2. Early DNA Methylation Variations Influence the Individual Susceptibility to Cancer and Instruct the Tumor Phenotype Following Homeostatic Rupture

In 2006, a review from Feinberg, Ohlsson and Henikoff formulated a provocative and unifying epigenetic model for cancer etiology to summarize the conclusions of their and other's extensive pioneering work. According to their model, epigenetic alterations should be expected to occur in a population of healthy stem cells very early during tumor development, even prior to the accumulation of specific tumor-initiating genetic alterations [32]. Residing stem cell populations possess the two key abilities defining stemness, i.e., self-renewal and multipotency. Self-renewal describes the capacity of a cell to maintain its own pool numerically unchanged throughout the entire life, whereas multipotency indicates the ability of undifferentiated cells to give rise to more than one mature cell type through a progressive commitment. Adult stem cell populations achieve these functions by different means, performing asymmetric or symmetric expansion and fate decisions [33] (schematized in Figure 1A). Satellite stem cells in the skeletal muscle perform apical-basal oriented cell divisions that generates daughter cells that are asymmetrically exposed to the stem-promoting niche [34]. Muscle stem cell growth and differentiation are therefore intrinsically determined by their asymmetric mode of division. By contrast, intestinal stem cells (ISCs) were elegantly proved to use symmetric cell division to maintain their pool. Neutral competition determines the location of daughter cells and their exposure to the niche-associated extrinsic cues regulating their commitment toward differentiation [35]. Sophisticated lineage-tracing and intravital imaging confirmed that cell division and cell fate decisions are uncoupled processes in ISCs [36]. Neural and epidermal stem cells were also observed to adopt symmetric division to replenish their pool and maintain the homeostasis of adult tissues [37,38]. As discussed in the next sections, DNA methylation dynamics control self-renewal and commitment of adult stem cells. The apparently unidirectional commitment of adult stem cells was challenged by findings indicating that committed progenitors can undergo dedifferentiation in order to replenish the stem cell compartment upon critical stress [39,40]. Importantly, oncogenic alterations increase in different manners the competitive fitness of mutated cells over their healthy counterparts, therefore biasing the competition that sustains homeostasis [41,42]. Likewise, early epigenetic changes would therefore result in the disruption of the homeostatic balance between stemness and differentiation. This imbalance would in turn increase the susceptibility of those progenitors to accumulate genetic alterations. Such epigenetic alterations are therefore likely to concern genes directly or indirectly involved in the maintenance of stemness. Stochastic, environmentally-induced epigenetic imbalance of stem cells would be followed by a cancer initiating hit involving tumor suppressors or oncogenes in the population of epigenetically disrupted progenitors. This genetic alteration further increases the genetic and epigenetic plasticity of the progeny, allowing the subsequent development of distinct subclones responsible for tumor evolution. The fact that loss of imprinting (LOI) of the *insulin-like growth factor II* (*IGF2*) gene in the pretumoral mucosa increases the risk of intestinal neoplasia formation upon loss-of-function of the *adenomatous polyposis coli* (*APC*) tumor suppressor probably represented the first convincing formal demonstration of this theory [43]. Multiple other lines of evidence seemed to support the model. First, some epigenetic features appear almost universal in human advanced neoplasia and are therefore considered to represent hallmarks of cancer, which has led to the general assumption that those

alterations occur very early during tumor development. These alterations include the hypomethylation of the genome in cancer cells, which can be accompanied by the focal hypermethylation of tumor suppressor genes. In addition, genomic instability that has been commonly used to explain rapid clonal evolution does not apply to all types of solid tumors, most of which are genomically stable and yet display high phenotypic plasticity, i.e., the ability to adapt to different challenges (e.g., multiresistance to treatment, ability to escape senescence and invasion of secondary tissues). This plasticity could represent the result of an early epigenetic imbalance [32].

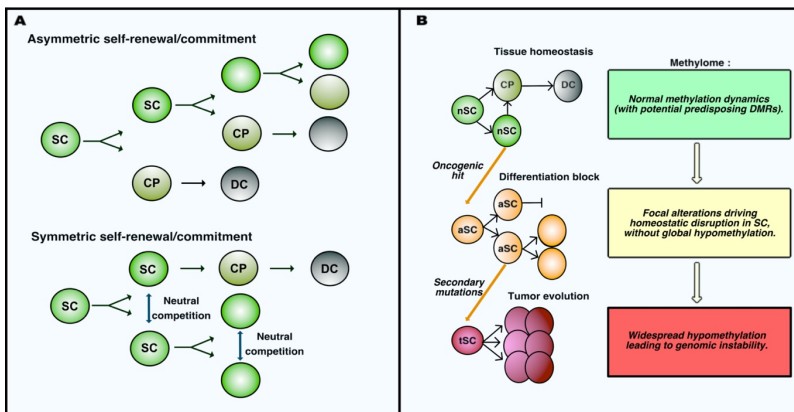

**Figure 1.** Adult stem cell dynamics during homeostasis and oncogenesis. (**A**) Self-renewal and commitment of adult stem cells occur via different modalities. In the case of asymmetric self-renewal, cell fate is determined by the polarity of stem cell (SC) division; indeed, only one daughter cell maintains its location in the stem niche allowing the maintenance of stemness, whereas the other is exposed to stimuli inducing the commitment into a cell progenitor (CP), which eventually maturates into differentiated cells (DC). In the case if symmetric self-renewal, daughter cells after division have equal chances for stemness and commitment, and cell fate is determined as the result of a neutral competition for maintaining interactions with the stem promoting niche. (**B**) Schematic representation of the initial steps of tumor development in parallel to the evolution of the methylation profiles in stem cells. Normal stem cells (nSC) physiologically perform self-renewal and generate committed progenitor cells (CP) that eventually mature into terminally differentiated cells (DC). Oncogenic alterations induce patterns of focal alteration of DNA methylation that impair the ability of altered stem cells (aSC) to differentiate, leading to their accumulation. These altered stem cells can accumulate further genetic/epigenetic mutations driving the tumor progression associated with widespread hypomethylation and instability of the cancer genome. Tumor stem cells (tSC) acquire malignant properties along this progression.

General hypomethylation of the genome represented the first widespread alteration initially described in colorectal cancer samples (CRC) compared to their surrounding healthy mucosa [44,45]. It later appeared as a universal feature of neoplasia, independently on the genetic pathway associated with cancer initiation [46,47], which led to the hypothesis that genomic hypomethylation may represent an early event in tumorigenesis. However, by characterizing the DNA methylation profile of intestinal stem cells (ISCs) in mice before and early after the inactivation of *APC*, a mutation associated with the initiation of 90% of CRC, we have recently shown that the first hit in the intestinal oncogenic sequence does not severely reduce the global methylation level of the genome in *APC*-deficient ISCs. Rather, this genetic alteration is associated with an early specific program affecting the DNA methylation of stem cells at defined loci involved in the control of their normal dynamics [48]. This was coherent with previous findings showing the absence of a general hypomethylation in the unstratified comparison between mouse adenomas and their normal surrounding mucosa [49]. Therefore, at least in the case of CRC, early oncogenic transformation does not result in a profound immediate rewiring of DNA methylation at genomic scale, which is characteristic of later stages of CRC. This notion implies a revised model in which early oncogenesis is accompanied by focal, rather than widespread

epigenetic changes determining the early phenotype of tumor stem cells. DNA methylation remodeling becomes more generalized along tumor progression, which in turn may be responsible for the genomic instability at more advanced stages of certain malignancies (Figure 1B).

Tumor initiation and development are also associated with the focal hypermethylation of specific regions that often correspond to tumor-suppressor genes, and several examples of hypermethylation occurring at various stages are known [50–54]. Remarkably, hypermethylation and silencing of certain loci are retrieved in nontumoral tissues in cancer patients and in the healthy mucosa surrounding adenomas in mice, suggesting that focal epimutations may occur even prior to genetic hits and represent the very first driver of transformation [55–57]. It seems therefore reasonable, at this stage, to affirm that focal epigenetic alterations can increase the susceptibility to cancer initiation or determine the early characteristics of tumor cells consequent to oncogenic hits. Long-lived cell populations involved in the long-term maintenance of the heathy tissue appear as the most prone candidates to be permissive for those alterations to exert a role [58]. The next section describes specific examples of the contribution of DNA methylation to stemness and commitment, and whether alterations in this epigenetic mark can bias the normal turnover of adult tissues at tumor initiation.

## 3. DNA Methylation Control of Cell Fate Decisions during Homeostasis and Disease: A New Paradigm of Tumorigenesis?

Like other epigenetic mechanisms, DNA methylation allows cells to achieve the expression of the appropriate networks of genes, in order to acquire and maintain specific identities depending on the environmental context in which they are embedded. Indeed, cell-type-specific methylation landscapes are associated with the different stages of development, spanning from totipotency of ESCs to maturity of adult differentiated cells [59,60]. In 2015, by combining the use of CRISPR/Cas9 genome editing and whole genome bisulfite-sequencing (WGBS), Liao and colleagues obtained the first comprehensive characterization of the specific role of each DNMT enzyme in the maintenance and differentiation potential of hESCs [61]. They found that, in contrast to what previously observed in mouse ESCs, DNMT1 inactivation is associated with global demethylation of the genome and is lethal in hESCs. Moreover, by characterizing overlapping and specific targets of *de novo* DNMT3A/B, they observed that their function is not essential for hESC maintenance and early development of the germ layers. This finding suggests that *de novo* methyltransferase activity is required for cell differentiation at later stages during development, when major modifications occur in the DNA methylation profiles. This conclusion seems coherent with the increasing amount of work demonstrating the importance of *de novo* DNA methylation dynamics during the commitment of adult stem/progenitors cells [4,59,60].

Multiple lines of evidences connect the plasticity of stem and tumor cells at epigenetic level. A pioneering work from the Jaenisch lab showed that nuclei from malignant cells transferred into enucleated oocytes undergo a reprogramming allowing cloned cells to revert to pluripotency and contribute to the formation of terminally differentiated tissues in chimeric mice [62]. This clearly suggested that epigenetic reprogramming coordinates the oscillation between pluripotent, differentiated and malignant cell state. Indeed, the progressive dedifferentiation observed during the progression of many human cancers has been associated with increasing reacquisition of embryonic stem cell-like DNA methylation profiles and cell identity [63,64]. Of note, a recent integrative pan cancer analysis conducted on thousands of tumor samples demonstrated that methylome signatures from several cancer types remain highly reminiscent of the DNA methylation profiles associated with the specific cells-of-origin from which malignancies arise [65]. This epigenetic memory in neoplasia supports the existence of an intimate epigenetic link between healthy adult stem cell and tumor properties. Importantly, several recent findings discussed in the next sections highlight the key role exerted by DNA methylation in orchestrating the homeostatic dynamics of adult stem cell populations. In parallel, accumulating evidence shows that aberrant DNA methylation patterns arise progressively in those populations, and set the stage for tumor establishment by impairing the balance between stemness and commitment in multiple different ways.

### 3.1. Hematopoietic Stem Cells

According to the well-accepted tree commitment model, all the cell types in the blood arise from a common pluripotent progenitor lineage, a hematopoietic stem cell (HSC), through a multistep process implicating a progressive reduction of lineage potential along the different steps [66]. This progressive commitment is associated with a stepwise modulation of gene expression patterns [67]. Indeed, HSCs can commit toward common myeloid or common lymphoid progenitor cells. Myeloid progenitor fate decisions can give rise to either megakaryocyte or granulocyte progenitors, and can determine whether lymphoid progenitors commit into more specific lymphoid precursors. This process terminates with the terminal maturation of megakaryocytes, monocytes and neutrophils, or CD4+ and CD8+ T-lymphocytes, natural killer (NK) cells and B-lymphocytes. Preliminary work has correlated such a remarkable adult developmental branching with the dynamic evolution of DNA methylation landscapes [68]. More recently, a single-cell-based characterization performed by Farlik and colleagues provided genome-wide stratified information about the DNA methylation dynamics along the different steps [69]. Importantly, these authors showed an interesting asymmetric inverse correlation in the extent of DNA methylation at regions containing the binding sites for TFs driving myeloid and lymphoid differentiation in the profiles obtained from the respective precursors. Such asymmetric DNA methylation landscapes constitute the foundation for cell fate decisions of hematopoietic lineages. Concerning the role of *de novo* DNMTs, DNTM3A activity was shown to be essential for the differentiation of HSCs, as its inactivation results in the accumulation of immature progenitors due to the hypomethylation and transcriptional upregulation of pluripotency factors associated with reduced expression of differentiation gene [70]. Remarkably, these findings were in line with the prevalence of DNMT3A mutations in myeloid malignancies [71,72] and leukemias [73], and provided a clear causal link with hematological oncogenesis. The same authors later showed that DNMT3B exert overlapping and specific functions to synergistically support DNMT3A during differentiation in hematopoiesis [74]. Previous evidence obtained in conditional mouse models showed that DNMT1 is also implicated in HSC commitment, with differentiation of myeloid-restricted progenitors being affected as a result of impaired gene expression upon adult conditional inactivation of this factor [75].

The lifelong multifate lineage commitment ability of adult HSC therefore represents a terrific example to illustrate the involvement of DNA methylation in homeostatic differentiation of adult progenitors, and whether an imbalance in its control can lead to tumor initiation via the unrestrained outgrowth of such populations as a result of a differentiation block.

### 3.2. Intestinal Stem Cells

The intestinal epithelium represents the fastest renewing structure in mammals, with the complete turnover of the tissue occurring in less than a week throughout the entire lifetime of an individual. This rapid turnover, functionally allowed by its simple architectural organization, make the intestinal epithelium a paradigmatic model for the study of adult stem cell properties [76]. Intestinal stem cells (ISCs) express the *bona fide* marker LGR5 [77] and are located at the bottom of intestinal crypts, intermingled between Paneth cells constituting their functional niche [76]. The workhorse LGR5+ ISCs constantly feed the transit amplifying (TA) cell compartment, located just above the stem-cell compartment and accounting for early-committed progenitors that can differentiate into the six mature secretory and absorptive cell types in the intestinal epithelium. It is now well accepted in the field that LGR5+ ISCs represent the primary cells-of-origin in colorectal cancer [78]. Although the prototypical role of ISCs in the study of epithelial stem cell biology, and the fact that CRC represented the first malignancy in which epigenetic alterations were reported [44], our knowledge on the epigenetic contribution to their function is still relatively limited. Important focal modifications have been observed in the methylation profiles during differentiation of epithelial cells [79]. Of note, these changes are not primarily associated with regions close to gene promoters but rather occur in active gene enhancers. Upon ISC commitment, a decrease in the methylation extent of certain

enhancers coordinates the binding of TFs promoting the expression of genes driving intestinal lineage specification. Conversely, enhancers of genes associated with stem identity become methylated, which allows the transcriptional silencing of those genes. In accordance with these observations, conditional deletion of maintenance DNMT1 in mice results in an expansion of intestinal crypts associated with a differentiation block of intestinal stem cells toward postmitotic lineages [79]. Interestingly, a previous report showed that the binding of TCF4, a TF critical for the homeostasis of the intestinal epithelium, shapes the epigenetic landscape in homeostatic ISCs by inducing the hypomethylation of specific enhancers, therefore modulating the expression of distant genes associated with stem cell functions [80]. The precise contribution of *de novo* DNMT3A and DNMT3B to this process is currently unknown.

We recently performed a characterization of ISC dynamics upon the earliest stage of intestinal tumorigenesis, and found that oncogenic inactivation of the Wnt inhibitor *Apc* in ISCs of mice induce a profound rewiring of transcriptional profiles associated with a specific DNA methylation signature [48]. Indeed, discrete epigenetic alterations occur at this stage, without the extensive remodeling of the DNA methylation landscapes that is characteristic of the later stages of CRC. Interestingly, we found that such alterations impair the responsiveness of ISCs to the signaling pathways governing differentiation, therefore resulting in the accumulation of proliferative $APC^{KO}$ ISCs at the expense of homeostatic commitment (Figure 2). Indeed, inhibiting *de novo* methyltransfersases preserves the homeostatic responsiveness to Wnt and BMP stimuli and the normal proliferation-to-differentiation dynamics in $APC^{KO}$ ISCs of intestinal organotypic models. *APC* plays therefore its important role of gatekeeper of the ability of stem cells to differentiate through a program coordinated by *de novo* DNA methylation. This was in line with the conclusions of previous studies showing an impact played by DNMT3A and DNMT3B enzymes in promoting intestinal tumor initiation in mice [81,82].

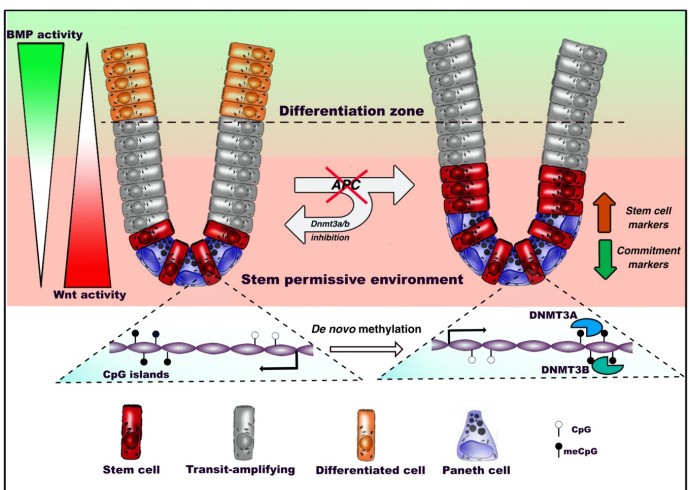

**Figure 2.** DNA methylation dynamics in adult intestinal stem cells (ISCs) during homeostasis and tumorigenesis. Under homeostatic conditions, intestinal stem cells residing at the bottom of the crypt between Paneth cells, where Wnt signaling is predominant, start migrating upward, proliferate intensively and progressively commit toward terminal differentiation. Oncogenic inactivation of *APC* biases *de novo* DNA methylation and reduces the ability of intestinal stem cells to respond to extrinsic bone morphogenetic protein (BMP) differentiation stimuli. This inhibits the expression of genes involved in the epithelial commitment, resulting in the outgrowth of the stem compartment. *De novo* DNA methyltransferases 3A (DNMT3A) and DNMT3B allow the occurrence of altered patterns, as inhibition of these factors preserves the ability of *APC*-null ISCs to differentiate.

## 3.3. Epidermal Stem Cells

Epidermal stem cells (EpSCs) ensure the maintenance of adult skin renewal throughout adult life through a perpetual cycle of growth and replacement, and also participate in the repair of the tissue after injuries. Mature epidermis consists in a stratified squamous epithelium in which only the basal

layer is mitotically active, and produces the extracellular matrix (ECM) constituting the underlying basement membrane that separates the epidermis from the dermis. As daughter cells leave the basal layer and migrate toward the skin surface, they withdraw the cell cycle, switch off ECM expression, and execute terminal differentiation [83]. Important remodeling of molecular and gene expression profiles occurs along this process. DNA methylation extent was found to vary during the different steps of differentiation, dynamically controlling the expression of cell-lineage-specific markers (Figure 3). Indeed, gene-regulatory elements associated with the differentiation of specific lineages are gradually demethylated during the transition from stem cells to terminally differentiated cells, and regions associated with inappropriate cell lineages or multipotency are increasingly hypermethylated during adult stem cell differentiation [84]. More recently, the dynamics and roles of *de novo* DNMT3A and DNMT3B were described in EpSC. During differentiation, these two enzymes decorate distinct enhancer and super-enhancer regions, and positively regulate the expression of self-renewal or differentiation markers in a dynamic manner via nonoverlapping mechanisms implicating the remodeling of the 3D chromatin architecture [85]. Indeed, Rinaldi and colleagues found a progressively increasing number of genomic sites occupied by DNMT3A during commitment, and a reduction of those occupied by DNMT3B, mirroring the expression of the two enzymes during differentiation. The same group later showed that both enzymes play important and unanticipated roles during tumor initiation and progression toward squamous carcinoma in mice [86]. Importantly, they found that Dnmt3a inactivation by itself does not exert any obvious impact on the homeostasis of the adult tissue or tumor initiation in mice. Rather, this lesion dramatically increases tumor burden upon chemically induced carcinogenesis. This is in part explained by the aberrant ectopic expression of genes involved in lipid metabolism consequent to DNMT3A loss-of-function. Moreover, they showed that Dnmt3b loss does not accelerate tumor initiation, but synergistically cooperates with DNMT3A loss during development toward more aggressive and metastatic stages of the malignancy. Together, these findings highlight important gate-keeping roles played by *de novo* methyltransferases during homeostatic commitment and multistep tumor initiation and progression in adult EpSCs.

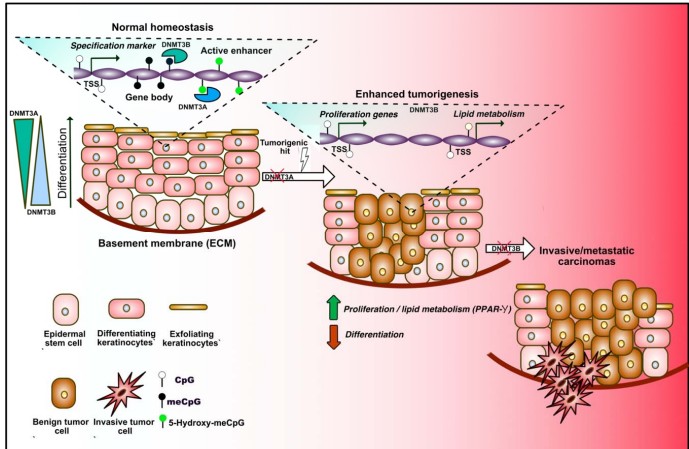

**Figure 3.** DNA methylation dynamics in adult epidermal stem cells (EpSCs) during homeostasis and tumorigenesis. Under homeostatic conditions, epidermal stem cells proliferate to renew the skin, constituted by different layers of differentiating keratinocytes (left in the sequence). Progressive changes in the DNA methylation profiles occur during differentiation. DNMT3A and DNMT3B binding dynamically alters the expression of lineage-specific markers by decorating transcription start sites (TSSs), gene bodies and enhancers. DNMT3A mediates 5-hydroxymethylation of active enhancers, therefore positively regulating gene expression. DNMT3A exerts a tumor suppressive role, since its loss increases adenoma formation upon carcinogenic hits, which is mediated by unrestrained expression of proliferation- and lipid metabolism-related genes (center). Loss of DNMT3B promotes the tumoral progression toward a more invasive stage (right).

*3.4. Neural Stem Cells*

Even after brain development is completed, a restricted number of neural progenitor/stem cells (NSCs) able to perform self-renewal persist in the adult brain and contribute to its complex functions by daily production of new neurons within two specific niches, the subgranular zone (SGZ) of the hippocampal dentate gyrus and the subventricular zone (SVZ) of the lateral ventricle [87]. The increasing knowledge with respect to the epigenetic networks regulating the complex maintenance and maturation of neural progenitors during neurogenesis has been reviewed by others [88,89]. Although DNA methylation profiling was largely performed in the brain, our understanding of the role of DNA methylation in the dynamics of NSCs remains largely fragmentary. DNMT1 is highly expressed by NSC precursors, and its loss-of-function reduces the number of neurons generated from those cells, although its inactivation does not affect NSC proliferation or differentiation, therefore suggesting that maintenance of DNA methylation patterns is critical for the survival of newly-formed neurons [90]. DNMT3A was reported to activate the expression of neurogenic genes via the methylation of intergenic and gene body regions, which negatively regulates the occupancy of the polycomb repressive complex 2 (PRC2) and the deposition of repressive trimethylation on the lysine 27 of the histone H3 (H3K27me3) [91]. Moreover, important differences in the DNA methylation patterns distinguish neuronal subtypes, therefore supporting a role of specific *de novo* patterns arising during cell type specification in the brain [92]. Interestingly, these changes seem reminiscent of the dynamics observed during embryonic development of the brain. Indeed, an increase of DNA methylation extent in the gene bodies of activated neuronal-related genes occur along differentiation of the embryonic mouse brain, and this gain is often accompanied by the loss of H3K27me3 during neuronal differentiation [93].

Although extensive profiling of DNA methylation in multiple brain tumors has recently been achieved, the functional implications of its dynamics in brain oncogenesis await further elucidation due to the lack of an appropriate cell type-specific data stratification. However, some interesting information regarding the epigenetic control of oncogenesis in the brain recently raised from the characterization of pediatric malignancies, and is documented in the next section.

## 4. Aberrant DNA Methylation as an Oncogenic Driver: Lessons from Pediatric Malignancies

During the past decade, cancer treatment decisions have been largely driven by the genomic profile of the patient's tumor. This "precision medicine" strategy has been effective in the management of many types of cancer. However, this increasing pan genomic characterization of neoplasia has shown important differences between adult and pediatric tumors. Indeed, the vast majority of pediatric malignancies harbor peculiar and limited genetic alterations compared to their adult counterparts [94]. This concept is particularly relevant in clinics, since targeting specific molecular pathways that are disrupted in adult malignancies frequently failed in ameliorating the prognosis of pediatric patients [95]. Importantly, the relatively low mutational rate characterizing pediatric tumors is associated with a remarkable dysregulation in their epigenetic landscapes [96]. Indeed, several tumor types display mutations and rearrangement of genes encoding epigenetic regulators. Among the most cited examples of epigenetic factors affected by these alterations are the subunits of the SWItch/Sucrose Non-Fermentable (SWI/SNF) chromatin remodeling complex in aggressive rhabdoid sarcomas [97,98], histone modifiers in leukemia [99,100] and histone H3 variants in pediatric diffuse gliomas [101,102].

Aberrant DNA methylation has proved its involvement in the oncogenesis of pediatric tumors. LOI of *IGF2* represents a frequent defect occurring in Wilms tumors, a form of kidney cancer that primarily develops in children. The oncogenic impact of this epimutation is associated with aberrant biallelic expression of *IGF2* associated with *H19* silencing, and was shown to represent an early predisposing alteration in the oncogenic sequence [103,104]. In a recent report, Saghafinia and colleagues showed that hypomethylation of the *MYCN* oncogene represents a universal feature of Wilms tumors [105]. In addition, they found that enhanced DNA methylation instability and hypermethylation of a specific subset of tumor suppressors are significantly associated with the subgroup of patients with the most unfavorable prognosis. DNA methylation profiling has also

been proved to be particularly meaningful for classifying pediatric tumors, in accordance with their genetic profiles and clinical outcome [106–109]. Interestingly, a recent stratification performed on cohorts of pediatric high grade gliomas (pHGGs) patients by Castel and colleagues, revealed that H3K27M-mutant diffuse midline gliomas (DMGs) display clearly different profiles with respect to all other pHGGs [110]. DNA methylation-based subclassification of DMG patients was concordant with the identity of the histone H3-variant oncogenic mutations associated with the different subgroups. This observation prompted the authors to suggest that specific methylation profiles in the different genetic subgroups may be reminiscent of the identities of tumor-cells-of-origin. The hypothesis is particularly relevant in the field, since tumor-initiating-cell populations are still unidentified in pHGGs and DMGs.

## 5. Concluding Remarks, Current Clinical Implications and Future Perspectives

I described here the recent remarkable improvement in our conception of the role of DNA methylation in homeostasis and cancer, which highlighted a significant contribution of epigenetic control to the transition between the two states. Almost fifteen years after the formulation of the "epigenetic progenitor model" of cancer, the conceptually fascinating idea that epigenetic alterations may modulate the relative susceptibility and orchestrate cancer initiation, seems more than ever relevant [32]. However, in the light of the recent findings discussed here, some important reassessment of certain aspects of the original model is necessary. First, it seems evident that some epigenetic alterations may occur in the healthy tissues, even prior to oncogenic mutations. However, it is important to verify whether those alterations occur in relevant tumor-initiating cell populations as well as their causal relationship with tumor initiation. In addition, widespread alterations, like genomic hypomethylation, were initially claimed as universal features of tumor cells, possibly occurring and exerting an impact before the genetically-driven cell transformation. However, this assumption is inconsistent with the recent characterization of DNA methylation profiling of the very initial stages of oncogenesis, which is associated with more focal and functional remodeling of DNA methylation patterns [48,85]. As supported by many studies cited in this article, epigenetic mechanisms, and DNA methylation in particular, have been more effectively associated with cancer initiation as a downstream effectors of the program imposed by oncogenic mutations. Indeed, altered DNA methylation dynamics have proved clear implications in early homeostatic impairment. Intriguingly, *de novo* DNMTs exert pleiotropic functions in adult stem cells, either by restraining (e.g., epidermis) or promoting (intestinal epithelium) tumor development upon oncogenic events. Mutations in *de novo* methyltrasferases may even represent the actual oncogenic hit, as observed in hematopoietic progenitors [70,74]. Ultimately, these findings corroborate the role of DNA methylation as a key gatekeeper of normal homeostasis in adult stem populations, and the increasing focus on those cells has provided a more meaningful functional stratification of the information in healthy tissues and tumors. This latter point will certainly benefit of the rapidly developing single-cell NGS technologies, which could inform us about the specific populations in which DNA methylation alterations first occur, and on the precise causal relationships with homeostatic ruptures and tumorigenesis.

Due to the control exerted on the function of oncogenes, tumor-suppressors and on genomic stability, DNA methylation is considered as an important source of diagnostic and prognostic markers, as well as a target for the development of therapeutic approaches [109]. Many aberrations in DNA methylation profiles in tumor samples are indeed included in the criteria for precise molecular diagnosis and classification, and used as prognostic biomarkers. This was extensively reviewed by others [111,112]. In addition, DNA demethylating agents are among the oldest and most widely used epigenetic drugs, whose effectiveness is conceptually based on their capacity to decrease the hypermethylation of tumor suppressors occurring in many tumors. Azaciditine (5-azacytidine) and decitabine (5-azadeoxicytidine) are cytidine analogues acting as DNA methyltransferase inhibitors representing the first FDA-approved demethylating agents for the treatment of myelodysplastic syndromes (MDS) and other hematological malignancies [113,114]. Their efficacy has also been evaluated by hundreds of preclinical studies

and tens of clinical trials on many types of solid tumors [115]. However, together with their significant toxicity, the precise antitumoral mechanisms of these agents remains debated, as their effect likely exceeds the expected reactivation of tumor-suppressors [116]. The accumulating knowledge on the role of DNA methylation in normal homeostasis of stem/progenitors and its alterations in tumor cells-of-origins may provide further significant translational insights. Therefore, I presented different examples of the functional impact exerted by *de novo* methylation and methyltransferases in both promoting or suppressing initiation of tumor development. In the case of CRC initiation, for example, DNA methylation remodeling in ISCs acts downstream to the oncogenic loss-of-function of *APC*. This affects the ability of ISCs to perform steady-state differentiation, which results in aberrant accumulation of premalignant stem cells, representing the basement for consequent tumor evolution. It has been shown that restoring the function of *APC* can revert CRC cells at any stage of tumor progression, even the most invasive, to normal cells, therefore re-establishing the normal crypt-villus homeostasis and promoting cancer regression without relapse [117]. The fact that reversible epigenetic mechanisms orchestrate the program exerted by this genetic alteration implies, in principle, the intriguing opportunity to recover normal homeostasis and achieve tumor regression by inhibiting *de novo* DNA methylation patterns in tumor stem cells. Interestingly, the treatment of $APC^{KO}$ adenomatous organoids with the DNMT3B specific inhibitor nanaomycin-A [118] was sufficient to restore the homeostatic balance of ISCs [48]. Future investigation is required to verify whether alterations remain reversible and such epigenetic strategy may be effective at more advanced stages of the disease, to force the differentiation of tumor stem cells and reduce their increased fitness over their healthy counterpart. In addition, epigenome-editing tools, such as TALE- or CRISPR-based DNA methylation editing promise exciting opportunities to dissect causal impacts and recover normal DNA methylation in cancer cells [21,119], as recently suggested in the case of neurological disorders [120].

Further investigation will therefore certainly extend our comprehension, and provide researchers with unanticipated vulnerabilities to target tumor stem cells.

**Funding:** This research received no external funding. At the time of conception of this work, M.B. was supported by the French Ligue Nationale contre le Cancer.

**Acknowledgments:** The author would like to acknowledge P. Jay, J. Pannequin (Institute of Functional Genomics, Montpellier) and M. Weber (Biotechnology and Cell Signaling, University of Strasbourg) for the discussions on stem cell biology and DNA methylation from which this manuscript "stemmed", F. Gerbe (Institute of Functional Genomics, Montpellier) for his insights and support in figure editing, and D. Castel (Gustave Roussy Cancer Institute, Villejuif) for his important inputs to the manuscript. I apologize to all colleagues whose work could not be cited due to space limitations.

**Conflicts of Interest:** The author declares no conflict of interest. The funders had no role in the design of the study; in the collection, analyses, or interpretation of data; in the writing of the manuscript, or in the decision to publish the results.

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
