# Peer review of "The Epigenetic Progenitor Origin of Cancer Reassessed: DNA Methylation Brings Balance to the Stem Force"

_2075-4655, 2020_

Round 1

Reviewer 1 Report

In this review manuscript entitled: “The epigenetic progenitor origin of cancer reassessed: DNA methylation brings balance to the stem force” the author provides an overview of the role of DNA methylation in cancer progression, with a special attention to its role on the maintenance of adult cell stemness. After introducing the connection between genetic alterations and epigenetic modifications in cancer, with particular emphasis on DNA methylation, he analyzes their interplay throughout the progression of the neoplastic event. Finally, he focuses on the dynamics of DNA methylation in some types of adult stem cells (i.e. hematopoietic, intestinal, epidermal and neural) and its control on tissue homeostasis, as well as its role in cancer progression. The central concept of the review consists in the temporal dynamics of the alterations in the DNA methylation landscape during cancer progression and their influence on genetic alterations. This is presented in a clear manner and it is reinforced with well-documented examples taken from the literature. The two figures are detailed enough and self-explanatory.

Minor points:

  • Line 30: reference needed

  • Line 55: In order to provide a more detailed picture of DNA methylation, the author should mention that CpG dinucleotide is underrepresented inside the genome but instead it accumulates at CpG island. Moreover, the author should mention (with references) some important key features of CpG island as the average number per gene, the mean distance from the TSS and specify which kind of genes have evolved the presence CpG islands within their promoter regions, compared to the genes lacking them.

  • Line 64: Please mention that CpG methylation can exist also inside gene bodies and it correlates with their transcriptional output (Baubec at al., 2015). This information is important also introduce the reader to the DNMT3A and PRC2 antagonism mentioned in paragraph 3.4

  • Line 77: reference needed

  • Line 80: reference needed

  • Line 89: reference needed

  • Line 118: “global hypomethylation of the genome in cancer cells, accompanied by the focal…”

  • Line 230 reference needed

  • Line 291: “progenitor”

  • Line 363: “have”

  • Line 367: The author should suggest what could be an appropriate experimental approach to discriminate between cancer progression events in which focal or global epigenetic modifications are a consequence of previous genetic alterations and cases in which epigenetic modifications are upstream and probably they are one of the causes of further genetic mutations.

  • Figure 2: mention that DMR stands for differentially methylated region

  • Figure 2: change “stem markers” with “stem cell markers” or “stemness markers” 

Author Response

Response to Reviewer 1 Comments

In this review manuscript entitled: “The epigenetic progenitor origin of cancer reassessed: DNA methylation brings balance to the stem force” the author provides an overview of the role of DNA methylation in cancer progression, with a special attention to its role on the maintenance of adult cell stemness. After introducing the connection between genetic alterations and epigenetic modifications in cancer, with particular emphasis on DNA methylation, he analyzes their interplay throughout the progression of the neoplastic event. Finally, he focuses on the dynamics of DNA methylation in some types of adult stem cells (i.e. hematopoietic, intestinal, epidermal and neural) and its control on tissue homeostasis, as well as its role in cancer progression. The central concept of the review consists in the temporal dynamics of the alterations in the DNA methylation landscape during cancer progression and their influence on genetic alterations. This is presented in a clear manner and it is reinforced with well-documented examples taken from the literature. The two figures are detailed enough and self-explanatory

I would like to thank this Reviewer for his positive evaluation of the manuscript, and for all his attentive suggestions to improve scientific relevance of the article. My answers to specific points are listed below.

  • Line 30: reference needed.
  • Please find this reference (1) at line 30 in the current version.

  • Line 55: In order to provide a more detailed picture of DNA methylation, the author should mention that CpG dinucleotide is underrepresented inside the genome but instead it accumulates at CpG island. Moreover, the author should mention (with references) some important key features of CpG island as the average number per gene, the mean distance from the TSS and specify which kind of genes have evolved the presence CpG islands within their promoter regions, compared to the genes lacking them.
  • Thanks for this attentive comment. Starting from line 64 and line 74 of the current version, please find extended information on the genomic distribution of CpG, CGIs, and their relationship with genomic elements related to regulation of gene expression (enhancers and intergenic regions). Please also find, starting from line 77, more relevant information regarding the functions of CGI and the co-evolution of promoters and CGIs. Please note that more detailed information regarding the genomic distribution of CGIs have been extensively reviewed by some of the authors cited in this review (please see doi:10.1038/nature08514, doi:10.1016/j.ceb.2007.04.011, doi:10.1016/j.jmb.2017.02.008 and doi:10.1038/nature07107). I would therefore prefer to avoid an in-deep description of these aspects that, although very important, are not essential for the focus of the manuscript

  • Line 64: Please mention that CpG methylation can exist also inside gene bodies and it correlates with their transcriptional output (Baubec at al., 2015). This information is important also introduce the reader to the DNMT3A and PRC2 antagonism mentioned in paragraph 3.4
  • Thanks for this important suggestion. The correlation existing between gene expression and DNA methylation extent at gene bodies is now illustrated at line 74.

  • Line 77: reference needed
  • Please find, at line 93 of the revised version, the appropriate reference (30).

  • Line 80: reference needed.
  • Please find, at line 54 of the revised version, the appropriate reference (14).

  • Line 89: reference needed.
  • Please find, at line 98 of the revised version, the appropriate reference (31).

  • Line 118;: “global hypomethylation of the genome in cancer cells, accompanied by the focal…”.
  • Please find this modification at line 147 of the current manuscript version.
  • Line 230 reference needed
  • Please find, at line 273 of the revised version, the appropriate reference (76).

  • Line 291: “progenitor”.
  • The word “progeniture” was meant to indicate “daughter cells” and not progenitor cells. In order to improve the clarity of the sentence, “progenitures”  was replaced by “daughter cells” at line 324 of the revised version.

  • Line 363: “have”
  • The subject of this sentence is represented by “epigenetic mechanisms” and not by “DNA methylation” (please see line 439). Indeed, I believe that the spelling can be considered correct.

  • Line 367: The author should suggest what could be an appropriate experimental approach to discriminate between cancer progression events in which focal or global epigenetic modifications are a consequence of previous genetic alterations and cases in which epigenetic modifications are upstream and probably they are one of the causes of further genetic mutations.
  • Thanks for this important suggestion. At line 448 of the current manuscript version, I now indicate single-cell next generation sequencing analysis as a tool allowing an improved chronological and functional interpretation of the cross-talk between genetic and epigenetic alterations that orchestrate cancer initiation and progression.

  • Figure 2: mention that DMR stands for differentially methylated region
  • Please note that this panel has been removed by the figure. The revised figure 1 in the current version illustrate the dynamics of adult stem cells and the contribution of DNA methylation during homeostasis and oncogenesis. This made the previous panel 2A unnecessary.

  • Figure 2: change “stem markers” with “stem cell markers” or “stemness markers”.
  • Please find this modification in the current version of the Figure 2.

Reviewer 2 Report

Thank you for the opportunity to review “The epigenetic progenitor origin of cancer reassessed: DNA methylation brings balance to the stem force” by Bruschi. This review article details the role of DNA methylation in the maintenance and acquisition of “stemness” as well as how alterations in DNA methylation perturb normal differentiation pathways, setting the stage for carcinogenesis. The author provides an overview of how DNA methylation impacts gene regulation, how DNA methylation marks are laid down and removed, and describes DNA methylation alterations in cancer. Examples of stem cells from four different tissues are provided: hematopoetic, intestinal, epidermal, and neural. Future directions, including how stemness-associated DNA methylation alterations can be targeted for cancer treatment, are discussed. Overall, this is an interesting and timely review that is well written and easy to follow.

I have some recommendations:

  • The reviewer centers around the concept of “stemness”, however, the properties that encompass “stemness” are not as clearly defined as they could be, particularly for individuals who are not experts in the field. The article would benefit from a more in depth description of what stemness entails and more clear definitions of terms used. For example, “self-renewal” and “genetic and epigenetic plasticity” are important concepts that are included could be explained in more detail. Introducing the concepts of symmetric and asymmetric self-renewal, and epigenetic relevance to these processes, could also help to provide further context for the phenomena depicted in Figure 2B.
  • The manuscript appropriately delineates the biology of embryonic stem cells and adult tissue stem cells. I believe that at an expanded discussion of how aggressive tumors reacquire an embryonic stem cell-like epigenetic state (for example, shown in Malta et al., Cell 2018 PMC5902191) would also be a useful addition to highlight how “stemness” is a continuum that can vary within and between tumors and is a major driver of tumor prognosis and outcomes.
  • It is unclear what the data presented in Figure 1A adds to the paper. There are not enough experimental details provided for the reader to assess what is being shown – for example, what technology was used for DNA methylation analysis in this experiment? The figure is blurry and difficult to read. The histograms suggest that the majority of sites are either very low or very highly methylated but the scatterplots suggest that there are a substantial number of sites that are methylated around 20 – 80% - I think that the density of points on these scatterplots is not being accurately conveyed. What the author is considering a “DMR” from these data are not clear. Are these DMRs consistent across samples or unique to an individual sample? These results may be better suited to include in a paper where you can give these findings the details that they deserve. I do like Panel B of this Figure.

Author Response

Response to Reviewer 2 Comments

Thank you for the opportunity to review “The epigenetic progenitor origin of cancer reassessed: DNA methylation brings balance to the stem force” by Bruschi. This review article details the role of DNA methylation in the maintenance and acquisition of “stemness” as well as how alterations in DNA methylation perturb normal differentiation pathways, setting the stage for carcinogenesis. The author provides an overview of how DNA methylation impacts gene regulation, how DNA methylation marks are laid down and removed, and describes DNA methylation alterations in cancer. Examples of stem cells from four different tissues are provided: hematopoetic, intestinal, epidermal, and neural. Future directions, including how stemness-associated DNA methylation alterations can be targeted for cancer treatment, are discussed. Overall, this is an interesting and timely review that is well written and easy to follow.

I thank this Reviewer for his attentive analysis of this work that has allowed me to enrich the general discussion according to his inputs. My answers to his specific points are detailed below.

I have some recommendations:

  • The reviewer centers around the concept of “stemness”, however, the properties that encompass “stemness” are not as clearly defined as they could be, particularly for individuals who are not experts in the field. The article would benefit from a more in depth description of what stemness entails and more clear definitions of terms used. For example, “self-renewal” and “genetic and epigenetic plasticity” are important concepts that are included could be explained in more detail. Introducing the concepts of symmetric and asymmetric self-renewal, and epigenetic relevance to these processes, could also help to provide further context for the phenomena depicted in Figure 2B.
  • Thanks for this important comment. I agree with this Reviewer on the fact that more extensive information regarding the normal dynamics of adult stem cells and the mechanisms related to their plasticity would provide the readership with a more didactic  overview of these populations. Starting from line 111 of the current version of the manuscript, please find the discussion regarding symmetric vs. asymmetric self-renewal and plasticity in the commitment of adult stem cells. These information are supported by examples related to different stem cell populations described in the literature. Please also see the updated version of the figure panel 1A, with an overview of symmetric and asymmetric stem cell dynamics.  Please note that other specific points related to the plasticity of the cross-talk between genetic and epigenetic networks are now discussed in the appropriate sections of the manuscript.

  • The manuscript appropriately delineates the biology of embryonic stem cells and adult tissue stem cells. I believe that at an expanded discussion of how aggressive tumors reacquire an embryonic stem cell-like epigenetic state (for example, shown in Malta et al., Cell 2018 PMC5902191) would also be a useful addition to highlight how “stemness” is a continuum that can vary within and between tumors and is a major driver of tumor prognosis and outcomes.
  • Thanks for making this important consideration on the role of epigenetic reprogramming in regulating the fluctuation of cells between pluripotent, committed and differentiated state. This is now discussed in the section 3 of the current version, starting from line 220, where some important related works are described.

  • It is unclear what the data presented in Figure 1A adds to the paper. There are not enough experimental details provided for the reader to assess what is being shown – for example, what technology was used for DNA methylation analysis in this experiment? The figure is blurry and difficult to read. The histograms suggest that the majority of sites are either very low or very highly methylated but the scatterplots suggest that there are a substantial number of sites that are methylated around 20 – 80% - I think that the density of points on these scatterplots is not being accurately conveyed. What the author is considering a “DMR” from these data are not clear. Are these DMRs consistent across samples or unique to an individual sample? These results may be better suited to include in a paper where you can give these findings the details that they deserve. I do like Panel B of this Figure.
  • Thanks for this analysis. The data displayed in the panel 1A were initially meant to support the conclusion that the earliest oncogenic lesion does not dramatically remodel the overall DNA methylation profile in tumor-initiating ISCs. However, I agree with this Review that this kind of representation is more appropriate in the context of an experimental article. These data can be found by the reader in the original reference, and are now simply discussed in the text. Panel 1A has been replaced with a schematic representation of adult stem cell fate dynamics.

Reviewer 3 Report

In this review, the author describes and discusses the role of DNA methylation in normal stem cell homeostasis as well as in cancer initiation. The topic of the review is very interesting and on time, however, some major and minor concerns should be addressed prior to publication:

Major concerns:

1- The majority of the sentences are too long and not well structured. For this reason, some concepts are difficult to understand. The author must be more concise avoiding repeating the same concept different times along the review. Therefore, extensive editing of the English language and style is absolutely required. 

2- Unlike adult malignancies, pediatric solid tumors show generally more epigenetic alterations rather than genetic mutations. Additionally, the DNA methylation alterations can be the driver of cancer initiation in some pediatric cancer. For example, the loss of imprinting of IGF2 is an early event of carcinogenesis in Wilms tumor (10.1016/j.canlet.2019.05.013). The author should include a new paragraph highlighting the possible different roles of DNA methylation alterations in adult and pediatric cancers. 

Minor concerns:

- It is useful to represent the figure 2A with images rather than with words.

- It can be useful to show the concepts written in the paragraph 3.3 (Epidermal stem cells) with a new figure. 

- Which are the authors refereed on line 302? it is better to substitute the word "authors" with Rinaldi and colleagues...

- The author writes the review discussion in plural form rather than singular. For example, he/she starts the discussion section with the word "we" rather than I on line 346. Why?

- It not clear to me which is the possible new paradigm of tumorigenesis showed in paragraph 3. The author should highlight this concept in this paragraph or alternatively he/she should change its title. 

Author Response

Response to Reviewer 3 Comments

In this review, the author describes and discusses the role of DNA methylation in normal stem cell homeostasis as well as in cancer initiation. The topic of the review is very interesting and on time, however, some major and minor concerns should be addressed prior to publication.

I thank this Reviewer for his fairly critical and constructive analysis of the manuscript. My answers to his specific concerns are listed below.

Major concerns:

  • The majority of the sentences are too long and not well structured. For this reason, some concepts are difficult to understand. The author must be more concise avoiding repeating the same concept different times along the review. Therefore, extensive editing of the English language and style is absolutely required. 
  • I agree with this Reviewer that the original version of the manuscript deserved an attentive editing of the English language. The current version underwent to an extensive editing, and a spelling check, in order to ameliorate the overall clarity of the content. Long sentences were simplified whenever possible, or spitted in multiple sub-statements. Please note that, for reasons of clarity, language modifications are not highlighted in yellow in the revised manuscript.

  • Unlike adult malignancies, pediatric solid tumors show generally more epigenetic alterations rather than genetic mutations. Additionally, the DNA methylation alterations can be the driver of cancer initiation in some pediatric cancer. For example, the loss of imprinting of IGF2 is an early event of carcinogenesis in Wilms tumor (10.1016/j.canlet.2019.05.013). The author should include a new paragraph highlighting the possible different roles of DNA methylation alterations in adult and pediatric cancers. 
  • Thanks for this important input. The original version of the manuscript was mainly focused on the role exerted by DNA methylation in controlling the dynamics of adult stem cells during homeostasis and oncogenesis. I agree with this Review that pediatric malignancies represent an important example of the role played by epigenetic alterations in oncogenesis. Please find these information in the new section 4, dedicated to pediatric malignancies. This section accounts for the discussion of multiple important references, including the one suggested by the Reviewer.

Minor concerns:

  • It is useful to represent the figure 2A with images rather than with words.
  • Please note that the information contained in the previous figure 2A were moved in the new version of the figure 1, which was reorganized following the inputs provided by Reviewer 2. Figure 1 now accounts for a more comprehensive illustration of the dynamics of adult stem cells, as well as the contribution of DNA methylation. The current version of figure 2 only accounts for a schematic representation of the dynamics of DNA methylation during homeostasis and oncogenesis in intestinal stem cells.

  • It can be useful to show the concepts written in the paragraph 3.3 (Epidermal stem cells) with a new figure. 
  • Thanks for this suggestion. Please find a schematic overview of what described in the section 3.3 in the new figure 3, added at the end of this section.

  • Which are the authors refereed on line 302? it is better to substitute the word "authors" with Rinaldi and colleagues...
  • Please find the required modification at line 336 of the revised manuscript.

  • The author writes the review discussion in plural form rather than singular. For example, he/she starts the discussion section with the word "we" rather than I on line 346. Why?
  • Thanks for making me notice this error. “We” was replaced with “I” wherever appropriate in the manuscript (highlighted).

  • It not clear to me which is the possible new paradigm of tumorigenesis showed in paragraph 3. The author should highlight this concept in this paragraph or alternatively he/she should change its title. 
  • Thanks for this important comment. I believe that the evidences described in this review imply some conceptual advances regarding the mechanisms that are functionally associated with oncogenesis, which overall deserve a provocative title. First, DNA methylation alterations appear as a common early feature associated with initiation of different tumor types, invariably resulting in the disruption of stemness/commitment balance, as suggested by multiple works that I cite. This is now clearly stated at line 232 in this section. Second, the timing at which alterations occur is different from what initially suggested by many (example: genome-wide hypomethylation does not represent an immediate feature of oncogenesis, which has some important implications). Last but not least, de novo DNA methylation dynamics appear essential for tumor initiation, as demonstrated by functional evaluation of the inactivation of de novo DNMTs (please see DOI: 10.7554/eLife.21697 and DOI:1158/0008-5472.CAN-19-2104). As proposed in the discussion, this implies that DNA methylation dynamics can represent an important vulnerability  to target, in order to recover homeostatic dynamics in tumor cells.

Round 2

Reviewer 3 Report

The author made a good job. He/she has successfully replied to all of my comments and the manuscript is now clear and convincing. On my side, it can be published in its present form.